# Greenwashing and Bluewashing in Black Friday-Related Sustainable Fashion Marketing on Instagram

Astrid Sailer [1], Harald Wilfing [1,*] and Eva Straus [2]

1 Evolutionary Anthropology, Human Ecology Research Group, University of Vienna, A-1030 Vienna, Austria; astridsailer@gmx.at
2 Occupational, Economic and Social Psychology, University of Vienna, A-1010 Vienna, Austria; eva.straus@univie.ac.at
* Correspondence: harald.wilfing@univie.ac.at; Tel.: +43-1-4277-54701

**Abstract:** Growing awareness of the fashion industry's negative impact on people and the environment has led to considerable growth of the sustainable fashion market. At the same time, Black Friday purchases increase annually as the sales event develops into a global phenomenon. As sustainable fashion brands are choosing to participate in the event, many communicate their offers via the social media platform Instagram. To gain a competitive advantage and maintain their sustainable corporate images, some brands use greenwashing and/or bluewashing strategies. The first part of this study explores which strategies were employed in Instagram content posted by sustainable brands, using quantitative and qualitative content analysis. We propose a research-based model of nine greenwashing/bluewashing strategies. The second part of the study examines predictive factors for consumer evaluations of Black Friday ads by sustainable brands, using an online survey and a stepwise multiple regression analysis. Findings show that consumers' critical attitude towards Black Friday and high ad skepticism predict positive evaluations while sustainable purchase behavior predicts negative evaluations. These insights suggest that 'sustainable' Black Friday campaigns may appeal to consumers who show a general concern for the environment and issues of social sustainability, but not to those who exhibit actual sustainable behavior.

**Keywords:** sustainability; marketing; greenwashing; bluewashing; sustainable fashion; social media advertising; Black Friday; consumer evaluation; green involvement; ad skepticism

## 1. Introduction

Public concern for the environment is rising. The Special Eurobarometer 464 survey reveals that the majority of Europeans (94%) have a personal interest in the protection of the environment. They are especially worried about climate change, air pollution, and waste production [1]. The fashion industry is a major contributor to environmental destruction. Textile production accounts for more greenhouse gas emissions than international aviation and maritime shipping combined [2,3]. Textiles equivalent to the volume of a garbage truck are sent to landfills or incinerated every second [2]. Additionally, the fashion sector is believed to be the second most impactful industry worldwide in terms of water pollution and consumption [4]. Synthetic textiles are the main source of primary microplastics in the oceans [5]. Growing awareness of the fashion industry's impact on the environment has led to a shift towards more sustainable options, resulting in a considerable growth of the global sustainable fashion market [6].

Brands recognize and seize the marketing opportunity that arises from society's environmental distress. There is an increase in the use of environmental appeals in advertising in times of acute environmental crises (e.g., nuclear disasters), suggesting that the rate of green advertising is linked to society's sentiment on environmental concerns [7,8]. The first peak of green claims in advertising was recorded in the 1970s followed by another spike in the 1990s [7,9]; but the use of green ads surged to a record high in the 21st century [8,9].

Exploiting the perks of environmental branding, some brands use exaggerated, deceptive, or unsubstantiated claims of environmental benefits in order to improve their corporate image [10,11]. This marketing practice, known as "greenwashing", has become an increasing issue. There has been an upsurge of misleading environmental appeals in advertising in the new millennium [8,9,12].

The prevalence of greenwashing has led to "growing confusion" [13] (p. 1886) and "alarming cynicism" [14] (p. 359) among consumers with regard to green advertising in general. Peattie and Crane [14] suspect the compartmentalization of green marketing—also referred to as "selective disclosure" [15]—as the core of the green advertising problem: as long as companies make green claims for merely one aspect of the supply chain instead of making the whole process transparent, there will be skepticism and concern among consumers [14].

Growing demand for transparency and accountability [16] has led to the establishment of Corporate Social Responsibility (CSR) programs and ESG (Environmental, Social, and Governance) criteria. Typical CSR programs communicate a company's intentions to tackle environmental and social issues [17]. Schaltegger and Hörisch [18] found that the main driver behind CSR programs is the aim to secure legitimacy rather than economic gains. Bansal and Clelland [19] demonstrated that merely expressing commitment boosts corporate legitimacy. Consequently, CSR claims are frequently exaggerated, selective, or simply unrealizable [20]. Similar to CSR, ESG policies are concerned with companies' environmental impact and social responsibility initiatives, but they also account for corporate governance [21,22]. ESG information is quantifiable, making CSR initiatives measurable and thus potentially valuable to stakeholders and investors. Data on ESG performance suggest that businesses profit from "investing in strong ESG policies" [21] (p. 23). Gillan, Hartzell, Koch, and Stark report a correlation between companies' returns on assets and their ESG scores [21]. Strong ESG performance is also associated with higher efficiency and firm value [21].

The practice of companies "paying lip service" to their CSR aims rather than taking substantial measures to improve their CSR performance is known as "bluewashing" [23] (p. 116). De Faro Adamson and Andrew [24] (p. 54) define this issue as follows: "a bluewashed company looks more socially responsible than it really is. Bluewashers see corporate social responsibility as a matter for the public relations department and do the minimum necessary to satisfy critics, advocacy groups, and social screens".

Given the benefit of developing and disclosing sustainability policies, many brands choose to communicate their initiatives via social media [25–27]. Minton et al. propose social media as the "ideal advertising medium for green advertisers" because interested users "self-select into sustainable lifestyle groups" [28] (p. 83). They also suggest that social media ads are potentially perceived as more credible than ads in traditional media due to the medium's dialogical character. On the basis of Moran, Muzellec, and Nolan's findings [29], Maslowska, Malthouse, and Collinger argue that brands are joining social media in order "to create engagement with not only their own customers but also with their customers' friends and followers" [30] (p. 469).

In terms of fashion, the social media app Instagram is among the most influential social media platforms, accounting for at least half of all brand posts according to an Exane BNP Paribas report from 2017 [26,31]. Influencers are a key factor in the app's success. The global market size of influencer marketing more than doubled in the last three years, growing to 13.8 billion U.S. dollars [32]. Recently, Instagram has adopted a shopping feature that enables businesses and creators to market and sell their products directly through the app [33]. The low-effort online shopping feature was introduced following extensive COVID-19-related restrictions of stores and shopping malls that led to an unprecedented surge of online shopping.

Even Black Friday, one of the busiest shopping days of the year [34], is increasingly taking place online: Black Friday e-commerce has seen a steady increase in the last decade [35]. In 2020, Black Friday online sales accounted for a record $9 billion spent in the US alone,

according to data acquired by Adobe Analytics [36]. These record numbers are in part owed to pandemic-related restrictions [35], but the trend is likely to continue. The worldwide spread of Black Friday as a result of globalization has led to an increase in international e-commerce [37].

Black Friday marks the beginning of the holiday shopping season and is one of the most important sales days of the year, particularly for retailers in the USA [34]. Apparel is among the most lucrative categories of Black Friday purchases in the US [38]. In 2018, 56% of online purchases from fashion retailers were made from mobile phones, reflecting a 6% increase from the previous year. Thus, mobile phones have become the "primary device of choice" for holiday shoppers [35]. In this regard, a connection to social media apps, such as Instagram, seems plausible. Although larger retailers have always recorded better sales performances on Black Friday than smaller retailers, Adobe Analytics found that the gap narrowed by over 200% as more consumers chose to support smaller businesses [36].

Black Friday sales are known for high discounts and great bargains. However, in recent years, critics have expressed doubts concerning the veracity of those deals, accusing retailers of applying "fictitious pricing", which is the practice of purporting "original" prices that are higher than the actual previous price [39] to create the illusion of high discounts. Others have been found to raise prices in the weeks prior to Black Friday. However, the majority of consumers believe in the veracity of discounts [40], which may explain the event's popularity. In addition, concerns have been voiced regarding the message so-called dumping prices convey, particularly in the fashion sector. As brands offer extreme discounts of up to 99%, both labor and goods are massively devalued [41]. Moreover, production volumes steeply rise around Black Friday, not only putting extreme pressure on garment workers but also leading to lay-offs and forcing factories to accept dumping contracts as order situations weaken afterward [42].

Nevertheless, the financial benefit of Black Friday has encouraged numerous supposedly sustainable fashion brands to participate in the sales event. Given the detrimental impact, Black Friday campaigns do not align with sustainability principles. In an attempt to maintain their positive corporate image, sustainable brands use greenwashing and/or bluewashing strategies in their social media communications to cloak their participation.

### 1.1. Literature Review

#### 1.1.1. Green Advertising

According to Schmidt and Donsbach [8], green advertising comprises all claims about the environmental benefits of a product, regardless of the claims' veracity. These claims may refer to the product or service offered, the company's production processes and other internal practices, or external measures, including donations and reforestation projects. Hartmann, Apaolaza Ibáñez, and Forcada Sainz [43] suggest that brands use environmental appeals in advertisements to implement a green brand identity. They distinguish between functional and emotional brand positioning strategies. Emotional strategies are used to evoke pleasant feelings in order to create positive associations with the brand. Such strategies may build on (1) the positive feelings associated with altruistic behavior [43,44], (2) the satisfaction experienced when displaying one's environmental consciousness through green products [43,45], or (3) the pleasant feelings experienced in natural environments [43,46]. Functional positioning strategies create positive brand associations by communicating environmentally friendly attributes of a product. Brands may, for instance, advertise environmental advantages over a competing product. A combined strategy of functional and emotional appeals achieves the strongest perceptual effects [43].

#### 1.1.2. Greenwashing

The term "greenwashing" was coined by environmentalist Jay Westerveld in 1986 and refers to the practice of making misleading claims about the environmental benefits of a brand or product. While it is not a new phenomenon, the rate of greenwashing has escalated in the new millennium as the demand for sustainable products increases [12].

In a study from 1991, Kangun et al. [10] identified misleading claims in 58% of all green ads in selected US magazines from 1989 to 1990. A study investigating green German and British print ads in the subsequent two decades (1993–2009) classified 77% of these ads as potentially deceptive [8]. In a report from 2009, the environmental marketing firm TerraChoice [47] found that 98% of products making environmental claims were guilty of at least one form of greenwashing. By 2010, the number of green products offered in North American stores had increased by 73%, while the proportion of greenwashing remained almost unchanged at over 95% [48]. As a basis for these studies, TerraChoice developed "the Seven Sins of Greenwashing", which are defined as follows [47] (p. 3):

1. The "Sin of the Hidden Trade-off" is committed by making environmental claims based on a very "narrow set of attributes" while disregarding other relevant aspects.
2. The "Sin of No Proof" is committed when brands provide no reliable evidence for their environmental claims.
3. The "Sin of Vagueness" is committed when brands use "poorly defined or broad" terminology to imply environmental compatibility, e.g., unregulated buzzwords.
4. The "Sin of Irrelevance" is committed when brands make claims that are not relevant for consumers seeking to make green purchase decisions (e.g., highlighting the absence of a harmful substance that is banned by law).
5. The "Sin of the Lesser of Two Evils" is committed by making environmental claims about a product that may be true in comparison to a competing product but disregard the negative environmental impact of the product category as a whole.
6. The "Sin of Fibbing" is committed whenever claims of environmental benefits are factually untrue or misleading.
7. The "Sin of Worshipping False Labels" is committed when brands use "fake labels", e.g., to imply third-party certification.

It must be noted that the aforementioned authors apply differing classification systems, making a direct comparison problematic. However, a growing tendency for greenwashing was identified by several authors [8,9,12], and the repeated expansion of greenwashing classification systems is reflective of the increasing trend as well. The inconsistency of these classification systems is one reason why greenwashing is such a difficult issue not only to research but also for consumers to detect.

A prominent example of greenwashing is the rebranding efforts by BP in response to a serious image problem. The oil and gas giant changed its name from British Petroleum to Beyond Petroleum and adopted a new green-and-yellow sun-like logo to falsely imply eco-friendliness [49]. Although BP's rebranding efforts were initially well-received and acclaimed for their "progressive and idealistic nature", they were soon dismissed as greenwashing due to the discrepancy between the green narrative and the company's actual environmental performance following the Deepwater Horizon scandal [50] (p. 577). According to Matejek and Goessling [50], green narratives are more willingly accepted when they do not digress too far from the established corporate image. Hence, rebranding is more easily detected as greenwashing when it clashes with the previous corporate image.

Recently, the greenwashing effect of celebrity endorsement has been the focus of substantial research. Credible celebrities—and even non-credible but attractive celebrities [51]—have a positive effect on the credibility of green claims [52]. Jin, Ryu, and Muqaddam regard Instagram marketing as an "evolution of celebrity endorsement" [53] (p. 666). They found that Instagram celebrities are perceived as more trustworthy than traditional celebrities in advertising [54]. It is, therefore, plausible that influencer endorsement has a strong greenwashing effect.

### 1.1.3. Bluewashing

The issue of bluewashing was first raised at the World Summit on Sustainable Development in 2002 in relation to the United Nations Global Compact (UNGC) (hence the color blue). Critics accused companies of using their UN partnership to conceal their poor enforcement of human rights and labor standards [55]. Corresponding research sub-

stantiates the claims of UNGC members shirking social responsibilities. Lim and Tsutsui provide indications of "ceremonial commitment" instead of substantive initiatives [56] (p. 79). Berliner and Prakash [23] discovered that members exhibit poorer performance than nonmembers in crucial and cost-intensive dimensions, while merely making low-cost efforts to improve in rather superficial dimensions.

Bluewashing is a fairly new phenomenon and research is rather difficult to discern due to a lack of standardized terminology and fuzzy boundaries. For example, some authors use alternative terminology, such as "corporate hypocrisy" [57] or "CSR-washing" [20], which, besides greenwashing and bluewashing, also covers other issues, including pinkwashing (in reference to breast cancer awareness) [58]. Other authors prefer the term "social-washing" [59] to clearly set the issue apart from environmental matters. Moreover, several authors regard bluewashing not as a separate strategy but as a constituent of its umbrella term greenwashing [60]. In this paper, the term bluewashing is used to refer to any misleading appeals about the social efforts or impact of a brand, product, or process [61].

A popular use of social appeals in advertising is cause-related marketing (CRM). CRM is defined as the practice of donating a portion of the proceeds from product sales to charitable causes [62,63]. According to Merz [64], this strategy achieves the best results when brands demonstrate long-term commitments to charitable organizations. Moreover, the donated amount must be appropriate in relation to purchase prices; otherwise, brands risk losing credibility [64]. However, CRM is frequently used to enhance corporate social images, while substantial dimensions, such as labor and human rights, are often neglected. In that case, CRM is used for bluewashing purposes.

### 1.1.4. Ethical Consumerism

Despite growing pressure on brands to align their practices with their policies, they shirk responsibility by claiming that it is the consumer's responsibility to make the right purchase decisions [65]. Building on the impact of consumer demand, they perpetuate the misconception that sustainable purchase decisions will ultimately regulate the market and pressure brands to implement environmentally friendly and socially responsible practices [65]. The idea behind this concept, which is generally referred to as 'ethical consumerism', is that consumers purchase not only a product but also the working conditions and practices to manufacture it [66]. Ethical consumerism thus frames consumption as a political act (cf. political consumerism) [67], drawing an analogy between sales receipts and voting ballots [65]. From this point of view, consumers cast a vote either in favor of or against people and the planet every time they spend money. Sustainable brands use this notion to persuade consumers to choose sustainable options over conventional ones in order to boost their sales. In effect, this idea of "consumer democracy" [65] cannot bring about fundamental change and can be considered greenwashing and/or bluewashing.

### 1.1.5. Home Economicus vs. Homo Sustinens

In 'orthodox economics', the concept of the so-called homo economicus is still predominant at present. This concept goes back to the term *economic man* coined by John Kells Ingram in "*A History of Political Economy*" [68]. In this highly simplified conception, a person is seen as a purely egoistic maximizer of their personal utility, a so-called rational agent. Since Garrett Hardin's study *The Tragedy of the Commons*, this conception has also found expression in resource economics or in the area of the management of so-called common goods [69]. Admittedly, this approach, subsequently, has not gone unchallenged. The broad studies of Elinor Ostrom may be mentioned here as an example [70].

As an explicit counter-model to homo economicus, the concept of homo sustinens coined by Bernd Siebenhüner and the counter-design of homo politicus by Malte Faber et al. can also be cited here [71,72]. The concepts of cognitive dissonance from social psychology [73] and the value–action gap concept from behavioral psychology [74], which are used in the context of sustainability discourses, also point to the fact that the concept of homo economicus is all too reductionist.

For our present study, these theoretical concepts are of importance insofar as they all stand for the fact that the broad variety of human behavior cannot be adequately explained by the model of homo economicus. For example, the very presence on the market of companies that produce textiles in a sustainable manner and charge a higher price for them can be described as a paradox. Homo economicus would not accept such products, whereas Siebenhüner's homo sustinens, for instance, would certainly take the aspects of sustainable production into account in their purchase decisions.

### 1.1.6. Green Consumers

Green consumers exhibit high green involvement [11]. According to Matthes, Wonneberger, and Schmuck, green involvement is reflected in "(1) environmental concern, (2) positive attitude towards green products, and (3) green purchase behavior" [13] (p. 1886). Highly involved consumers are aware of environmental issues and the necessity to protect the environment [13]. They have a positive attitude regarding "the advantages, favorability, or the quality of green products" [13] (p. 1887) and also choose environmentally friendly products over conventional ones [75], i.e., they exhibit no value–action gap [76]. Moreover, eco-relevant knowledge appears to be associated with green involvement [77].

### 1.1.7. Green Ad Skepticism

According to Matthes and Wonneberger [11], green consumers attribute higher informational utility to green ads, meaning that they find the information presented in green ads relevant to their purchase decisions. In contrast to previous faulty findings, Matthes and Wonneberger [11] found that skepticism towards green ads was lower in green consumers. However, they noticed that green consumers elaborate ads more closely. Investigating ad elaboration, the authors found that green consumers take longer periods of time to scrutinize ads. They suggest that if statements in ads are trustworthy and show "high informational utility", consumers will evaluate them positively. Conversely, "low informational utility" will lead to ad distrust [11]. Kießling [78] found that greenwashed ads have a lower persuasive effect on consumers who exhibit high green involvement (i.e., green consumers) than low-involvement consumers.

### 1.2. Research Focus

The significance of Instagram in fashion marketing and the role of social media in the communication of brands' sustainability intentions signal a necessity for research into sustainable fashion marketing on Instagram. While some recent studies have examined this field of interest [26,79], no existing research has focused on Black Friday-related content in this context.

There is also a research gap in the field of greenwashing and bluewashing research, which is predominantly concerned with non-sustainable businesses as opposed to sustainable SMEs (small and medium-sized enterprises). Furthermore, there are no prior studies focusing on consumer evaluation in this particular field of interest. This study contributes to a better understanding of the appeal of sustainability-centered advertising and provides valuable insight into sustainable brands.

For this purpose, we conducted a two-part study. In the first part of the study, we performed content analyses of Black Friday-related Instagram content posted by sustainable fashion brands and established a research-based model of greenwashing and bluewashing strategies. In the second part of the study, we conducted an online survey using a selection of the analyzed Instagram posts. Stepwise multiple regression analyses of the survey responses were performed to identify predictive factors for consumer evaluation. We formulated two research questions:

*Research question 1: Which greenwashing and bluewashing strategies can be found in Black Friday-related content posted on Instagram by sustainable fashion brands?*

*Research question 2: Which factors are the most relevant predictors of (a) brand evaluation and (b) sustainability evaluation by consumers?*

A model depicting the explorative research design of the second part of the study is shown in Figure 1. Prior research suggests that high green involvement has a negative effect on green ad persuasiveness [78]. As an extension of the concept of green involvement [13], we suggest the term sustainability involvement, which, in addition to environmental concerns, also includes social sustainability issues, such as social equity, human rights and labor rights issues, among others. The sustainability-conscious consumer thus exhibits high eco- and social consciousness, a positive attitude towards green and 'blue' (fair and ethically produced) products, and actual green and 'blue' (socially responsible) purchase behavior. Given that high involvement results in lower persuasiveness of greenwashed ads [78], we hypothesize that consumers who exhibit high sustainability involvement give negative brand and sustainability evaluations.

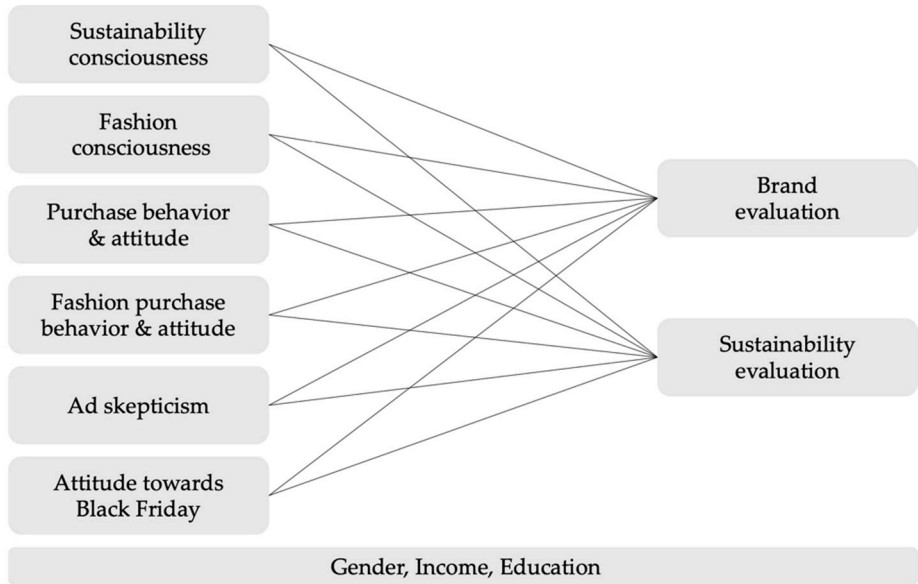

**Figure 1.** Model.

**Hypothesis 1 (H1).** *High sustainability involvement (i.e., high sustainability conscious-ness and sustainable purchase behavior) negatively affects consumer evaluation of Black Friday-related Instagram content displaying green and/or blue claims.*

## 2. Part 1: Content Analysis

### 2.1. Materials and Methods, Part 1

In the first part of the study, Instagram content related to the Black Friday sales event posted by sustainable fashion brands in 2020 was qualitatively analyzed for claims of social and/or environmental benefits. Only Instagram content in German or English was considered for this research. Data were collected using a variety of methods. Firstly, lists of top-rated sustainable fashion labels compiled by impartial third parties were con-sulted [80,81]. Instagram accounts of the ranked labels (76 brands in total) were scanned for eligible content posted around Black Friday 2020. Secondly, Instagram was screened using relevant, topic-specific hashtags, including #noblackfriday, #greenfriday2020, #greenfriday, #greenweek, #fairfriday, #fairweek, #socialfriday, #bluefriday, and #blueweek, since these hashtags are frequently used by sustainable brands. Thirdly, related content that was featured on Instagram as sponsored ads (what sort of sponsored ads Instagram users are presented with is based on the content they engage with; thus, if a user heavily engages (i.e., searches, saves, shares, likes) with content about sustainable fashion and Black Friday, the sponsored ads on their feed are likely to feature related content) during data collection as well as the corresponding Instagram accounts were scanned.

Qualitative content analysis of all eligible posts was performed as described by Mayring [82]. Visual (e.g., backdrops and separate images or videos), textual (e.g., text on

slides or in captions), and functional (e.g., hashtags and influencer tags) elements were analyzed. All claims of social and environmental benefits implied through linguistic and visual insinuations or intertextual references (i.e., content creator collaborations) were collected. An inductive approach was used to identify patterns for categorization [82,83]. Drawing on existing literature on greenwashing and green advertising [8,43,47,50,52,65], the identified green and blue claims were categorized. We adapted the greenwashing categories for the classification of bluewashing instances. All categories were then analyzed quantitively. Using IBM SPSS 27 statistical software, a Fisher's exact test was performed to identify a potential statistically significant difference between greenwashing and bluewashing occurrences within the categories. Lastly, where applicable, the categories were classified as emotional or functional positioning strategies, based on the brand positioning strategies described by Hartmann, Apaolaza Ibáñez, and Forcada Sainz [43].

### 2.2. Results, Part 1

In the first part of the study, greenwashing and bluewashing instances in Black Friday-related Instagram ads by sustainable brands were categorized based on prior research on greenwashing, which we adapted for bluewashed ads. We propose nine Black Friday-relevant categories (Table 1).

**Table 1.** Greenwashing and bluewashing categories.

| Category | Description | Example |
| --- | --- | --- |
| Cause-related marketing (CRM) [8,64] | Offering to make a small donation to a charity for each purchase | Reforestation project; 10% per order towards charity (e.g., Fashion Revolution) |
| Ethical consumerism argument [65,67] | Perpetuating the idea that it is the consumers' responsibility to make sustainable decisions rather than the brands' | Persuading customers to choose CRM options over conventional discounts |
| Promotional gift [43] | Offering a social or eco-gift with each purchase instead of discounts | Scrap material key chain, products with activist slogans |
| Sin of the Lesser of Two Evils [47] | Promoting a product or process that is less socially or environmentally harmful than a comparable one but still has a negative impact | Promoting Black Friday sales of sustainable fashion while condemning fast fashion brands for offering discounts |
| Sin of Fibbing [47] | Making misleading or false claims about the environmental or social benefit of a product or process | Claiming that consumers are saving resources when they purchase sustainable products |
| Sin of Vagueness [47] | Using buzzwords and unstandardized terminology to imply favorable environmental or social performance | Made with love; planet- and people-friendly fashion |
| Rebranding [50] | Changing the name of the sales event to imply improved social or environmental compatibility | Green Friday or Blue Friday instead of Black Friday |
| Content creator endorsement [52] | Using credible content creators to advertise a product or brand to enhance the credibility of sustainability claims | Greenfluencer * campaign, social activist endorsement |
| Imagery [43] | Using nature-related or charity-related imagery so that consumers associate the brand with the feelings evoked by what is depicted | Images of natural scenery or acts of charity |

* The term *greenfluencer* refers to content creators who educate and inform about environmental issues and often market green products.

Eligible Black Friday-related content was found on 39 Instagram accounts of sustainable fashion labels. In all relevant Black Friday-related posts, we found a total of 115 green- and bluewashing instances. Table 2 shows the number of occurrences of all strategies. Rebranding and social CRM were used rather frequently for bluewashing purposes, while other categories were not used at all (e.g., content creator endorsement and the ethical consumerism argument), or only in combination with greenwashing (e.g., the Sin of the Lesser of Two Evils). All categories were used for greenwashing purposes. Fisher's exact test indicated no significant difference in the frequencies of greenwashing and bluewashing instances within the categories ($p = 0.139$, FET).

**Table 2.** Quantitative data.

| Category | Green-Washing | Blue-Washing | Both Green- and Bluewashing | Total per Category |
|---|---|---|---|---|
| CRM (environmental/social) | 16 | 15 | 1 | 32 |
| Rebranding | 17 | 11 | - | 28 |
| Imagery (nature/charity) | 13 | 6 | - | 19 |
| Sin of Fibbing | 6 | 5 | 3 | 14 |
| Sin of the Lesser of Two Evils | 8 | - | 2 | 10 |
| Content creator endorsement | 5 | - | - | 5 |
| Gift (eco-/social) | 3 | 1 | - | 4 |
| Sin of Vagueness | 1 | 1 | - | 2 |
| Ethical consumerism | 1 | - | - | 1 |
| Total per sustainability focus | 70 | 39 | 6 | =115 |

Overall, the most frequent category was CRM, used by 32 out of 39 brands. CRM was often used in combination with the second most frequent category: rebranding. A combined strategy of the following three categories was most common: CRM, rebranding, and nature- or charity-related imagery (seven cases, plus eight times in combination with additional categories). All but three brands used combinations of several categories. Combinations of any three categories were most frequent (18 cases), followed by two and four combined categories (both eight cases). The highest number of combined categories used by one brand was six; these strategies were used for greenwashing purposes.

Drawing on Hartmann, Apaolaza Ibáñez, and Forcada Sainz's brand positioning strategies [43], three categories can be classified as emotional positioning strategies: CRM, the use of imagery, and promotional gifts. Another three categories can be classified as functional positioning strategies: the Sin of the Lesser of Two Evils, the Sin of Vagueness, and the Sin of Fibbing. Seventeen brands used a combination of both positioning strategies in their Black Friday campaigns. Eighteen brands used only emotional but no functional strategies. Four brands relied solely on functional strategies.

### 2.3. Discussion Part 1

In the first part of the study, we performed a qualitative content analysis of Black Friday-related Instagram posts by sustainable fashion brands. The aim was to identify strategies used to greenwash and/or bluewash the repercussions of Black Friday or to mask a brand's participation in the unsustainable sales event entirely, in order to maintain their sustainable corporate image.

We found a total of 115 greenwashing and/or bluewashing instances in Instagram content posted by 39 sustainable brands. To answer the first research question, we defined nine categories based on previous findings in research on greenwashing: CRM, the argument for ethical consumerism, promotional gifts, rebranding, the Sin of Vagueness, the Sin of Fibbing, the Sin of the Lesser of Two Evils, content creator endorsement, and the use of sustainability-related imagery.

Black Friday campaigns first and foremost aim to incentivize purchases, conventionally with the use of discounts. In the case of greenwashed and/or bluewashed campaigns, discounts are frequently substituted with other seemingly more sustainable offers.

The vast majority of brands used CRM strategies, offering to make small monetary or in-kind donations to a charity or reforestation project for each order. While CRM is not universally recognized as greenwashing (or bluewashing), Schmidt and Donsbach [8] mention donations, reforestation projects, and similar external measures as a form of green advertising. In addition, Merz [64] argues that CRM strategies are more likely to be credible when brands make long-term commitments. With regard to Black Friday, however, brands used short-term CRM campaigns to boost their sales while protecting their corporate images. Such performative acts of charity can be considered bluewashing or greenwashing. Drawing on Hartmann, Apaolaza Ibáñez, and Forcada Sainz [43], CRM strategies can be classified as an emotional positioning strategy built on the satisfaction evoked by altruistic behavior. Black Friday, which is commonly associated with overindulgence, is reframed as an altruistic act. Instead of the negative feelings related to the adverse effects of the sales event, the reframed purchasing act thus evokes positive emotions. Furthermore, CRM strategies imply that the success of the advertised charity project is contingent on the purchases of a brand's products. Hence, CRM has a great potential to incentivize consumption. In addition, Black Friday-related CRM appears to be less costly for brands, as conventional discounts usually exceed the amounts donated.

The financial benefit might be one reason why the ethical consumerism argument was used to persuade consumers to choose donation options over discounts. Brands strategically use this argument to delegate responsibility and accountability away from brands towards consumers. As described above, the argument builds on the misconception that consumers' sustainable purchase decisions have a lasting regulatory effect on the market, claiming that it is their responsibility to choose sustainable options over conventional ones in order to pressure brands to adopt more sustainable practices. As regards Black Friday, the argument was used so that brands could offer unsustainable discounts without laying themselves open to criticism: consumers were given the choice between a discount and an environmental CRM option.

With the aim of attaching a positive connotation to Black Friday, some brands substituted discounts for sustainability-related promotional gifts. While the purchase incentive remains, the risk of negative Black Friday-related connotations is minimized. For consumers, these gifts serve as an incentive, not only because they receive items 'for free' but also because they can use these items to show off their support for a charitable cause to others. This notion can be explained by one of the three emotional positioning strategies described by Hartmann, Apaolaza Ibáñez, and Forcada Sainz, which builds on the satisfaction of exhibiting one's sustainability consciousness through the "socially visible consumption" of sustainable goods [43,45] (p. 11). Examples of such gifts are special collection items designed to show support or raise awareness for charitable causes, for instance, through displays of activist slogans.

Like discounts, such donation campaigns and gifts are limited offers used intentionally to encourage fast purchases, which often result in overconsumption and impulse buys. Thus, while the outcome is similar, these strategies create a false impression of a more sustainable approach (or at least a skewed image).

Rebranding is used to achieve the same effect. The strategy is commonly used to form new positive associations for previously unfavorable brands or practices [50]. Sustainable brands used this tactic to reframe Black Friday as a people- and planet-friendly sales event. The most prominent greenwashing example is Green Friday, which was established as a countermodel to Black Friday. Bluewashing examples include Fair Friday and Social Friday. By changing the name of the sales event, sustainable brands can participate without the risk of being associated with the negative repercussions of Black Friday (e.g., price dumping and overconsumption). The strategy is used to imply that the rebranded version is fundamentally different from its unsustainable forerunner, even though it merely redirects consumption away from conventional brands towards sustainable brands. The rebranding strategy is particularly significant for the Instagram setting since it was also used in the form of hashtags which allow users to identify and access relevant posts.

The terms used to rebrand the sales event are often buzzwords, such as 'green' or 'fair', that are immediately associated with sustainability but are broad and unspecific. In these cases, brands make use of the Sin of Vagueness. To avoid double counts, rebranding instances were not counted towards the Sin of Vagueness in this study. Nevertheless, these brands enjoy the benefits of the Sin of Vagueness: since the terminology is unregulated, there is no risk of accusations of making false claims, while the positive connotation of sustainability is attached to the brand. The Sin of Vagueness also applies when brands make ambiguous statements. 'Made with love', for instance, implies that a product was manufactured by happy garment workers under great working conditions, yet no explicit claims are made.

Whenever such claims are factually untrue or misleading, brands are guilty of the Sin of Fibbing. A bluewashing example of this strategy is offering discounts on the pretext of accessibility, i.e., claiming to offer a discount so that lower-income consumers can afford the products, while maintaining high profit margins throughout the year [84]. Another example is the claim that price reductions merely affect profit margins even though they decrease the value of products and labor and are thus harmful to garment workers [41]. A common greenwashing example of the Sin of Fibbing is the claim that consumers are saving resources when they purchase sustainable products. Although a sustainable product might require fewer resources than a conventional one, the purchase of a new product can never save resources.

The Sin of the Lesser of Two Evils is most representative of the whole problem at hand. Sustainable brands aim to improve current conditions. While that is important, the underlying issue—environmental destruction due to the depletion of resources and the immense amount of textile waste as a direct consequence of overconsumption—cannot be solved by producing new products under improved conditions. The presented Black Friday campaigns might advertise more socially and environmentally compatible clothes, but they still encourage overconsumption and are thus just the lesser of two evils. For this study, we classified only those Black Friday ads as guilty of this Sin that explicitly made claims of sustainability due to improved conditions (e.g., bashing worse conditions of other brands). The three Sins are used to provide information on environmental or social benefits and can therefore be classified as functional positioning strategies [43].

Misleading claims are rendered more credible with the use of celebrity endorsement [52]. Since influencer marketing is a reformed version of this marketing strategy within the social media context [53], there is reason to assume that content creator endorsement also has a greenwashing effect. The effect is perhaps even stronger, given that influencers are perceived as more trustworthy than traditional celebrities [54]. As credible and authentic advocates, greenfluencers render the green claims made by the brands they endorse more trustworthy. Social activist influencer campaigns (e.g., feminist content creators or human rights activist influencers) are often used for bluewashing purposes but were not detected in this study.

Furthermore, since Instagram is a visual media app, content creators who focus on sustainability issues usually post related visual content. Greenfluencers' images, for instance, may depict natural scenery or nature-related elements. As described above, Hartmann, Apaolaza Ibáñez, and Forcada Sainz [43,46] categorize the use of pleasant nature-related imagery as an emotional positioning strategy. Such depictions of nature create positive associations between Black Friday and the environment because they build on the positive feelings people experience in natural environments. Similar positive feelings might be aroused by depictions of charity, such as displays of philanthropic acts (e.g., children receiving aid). Negative imagery, such as images of environmental destruction or humanitarian crises, for instance, might evoke feelings of compassion and incentivize consumers to 'help', i.e., to make a purchase.

Nearly all brands used a combination of several greenwashing and/or bluewashing categories—in some cases as many as five or six categories. Brands may benefit from

employing multiple strategies, as it is likely that consumers recognize only some of the strategies while others go unnoticed.

Based on Hartmann, Apaolaza Ibáñez, and Forcada Sainz's positioning strategies [43], the vast majority of the sustainable brands in this sample appealed to their customers' emotions and approximately half used functional appeals to persuade consumers. Nearly half the brands combined emotional and functional strategies, which, according to the authors, is most effective. These brands may have targeted a wide audience, perhaps aiming to persuade consumers exhibiting high involvement with the high informational utility of the ads [11] and to evoke emotions in low-involvement consumers.

By employing such a wide variety of greenwashing and bluewashing strategies, sustainable brands not only create a positive connotation for an unsustainable sales event but also legitimize the use of such deceptive tactics in all ads. When sustainable brands make greenwashing and bluewashing common practice, they enable conventional brands to do the same, making it more difficult for consumers to make sustainable purchase decisions.

## 3. Part 2: Survey Research

The second part of the study further extends the knowledge about greenwashing and bluewashing strategies in 'sustainable' Black Friday campaigns. The findings of the content analysis above constitute the basis for the survey research. This second part provides valuable insight into consumer perceptions of such content.

### 3.1. Materials and Methods, Part 2

For the survey research, we employed an explorative research design, testing the predictive ability of a number of factors for consumer evaluation of greenwashing and bluewashing in Black Friday ads. The quantitative approach allows us to draw objective and reliable conclusions about consumer attitudes and evaluations of sustainable fashion brands.

#### 3.1.1. Procedure and Sample

We conducted an online survey to examine consumer evaluations of Black Friday ads by sustainable fashion brands. A small number of relevant posts analyzed in the first part of the study were selected as stimuli. The survey was conducted via SoSciSurvey over a period of 40 days from 13 May–21 June 2021 and was accessible via PC/laptop and smartphone/tablet.

Participants ($n$ = 148) were recruited online via a URL link shared on multiple social media platforms (Instagram, Facebook, and LinkedIn). Participants were primarily German speakers; questionnaire items were phrased in German. Instagram content was left unaltered in English or German language. Only participants who can read and understand English language text and who are familiar with Instagram were authorized for participation. Questionnaires were considered for data processing provided that at least page 30 of the survey (the last page of brand evaluation in the PC version) had been completed. The resulting sample size was $n$ = 148, except for evaluations of brand 6, for which $n$ = 135. Average completion time of the survey was 21 min.

An optional quiz on the environmental and social repercussions of the fashion industry at the end of the survey was used as an incentive for participant recruitment. Questions were predominantly based on the 2020 Fashion Revolution White Paper [85] and some additional sources [86–91].

#### 3.1.2. Measured Constructs

The online survey was divided into three parts. The first part examined participants' sustainability-related attitudes and behavior. The question format was a six-point Likert scale (1 = strongly disagree, 6 = strongly agree). Formats without neutral options were used to avoid ambiguous responses which could indicate either conflict (i.e., 'the one as well as the other') or indifference (i.e., 'neither the one nor the other') [92,93]. Based on

the concept of green involvement [13], participants' general sustainability involvement was examined, including their green and 'blue' purchase behavior and attitude towards green and 'blue' products, as well as social and eco-consciousness (one item omitted to increase reliability). Additionally, participants' sustainable fashion-related involvement was examined (one item omitted to increase reliability). Participants' ad skepticism and attitudes towards Black Friday and Instagram were also examined.

The second part of the survey investigated consumer evaluations. Black Friday-related Instagram posts analyzed in the first part of the study served as a basis for this section. Participants were presented with three pairs of visual stimuli. Each pair consisted of one set of Instagram posts that predominantly used greenwashing tactics and one set that predominantly used bluewashing tactics. Each set featured two to three Instagram posts by one brand. The sets of each pair were roughly similar to each other in terms of the quality and quantity of employed greenwashing/bluewashing categories. The first set used CRM and rebranding. The second set used the Sin of the Lesser of Two Evils and the Sin of Fibbing. The brands of the third set each used at least four categories, including CRM, rebranding, imagery, and the Sin of the Lesser of Two Evils.

Using contrasting adjectives (e.g., ethical/unethical) on a six-point semantic differential scale [75] (1 = negative, 6 = positive), participants were asked to evaluate the brands based on the visual stimuli. This format was also used to examine participants' evaluations of the brands' sustainability-related performances. Six-point Likert scales were used to investigate whether participants suspected or recognized greenwashing and/or bluewashing in the presented ads; we refer to this scale as 'sustainability skepticism'. Participants' prior knowledge of the fashion brands as well as their purchase intention was also inquired about; these items were adapted from Knes [94]. Table 3 shows all relevant measured constructs, including example items.

**Table 3.** Measured constructs.

| Construct | Items | Example Item |
|---|---|---|
| Sustainability consciousness [92,94,95] | 7 | "The condition of the environment affects the quality of my life." |
| Purchase behavior and attitude [13,92,96] | 5 | "When I have a choice between two equal products, I purchase the one less harmful to other people and the environment." |
| Fashion consciousness [94] | 3 | Conventional fashion brands often do not care about the environment. |
| Fashion purchase behavior and attitude [94] | 6 | I am willing to switch to other brands if they produce clothing under ethical and environmentally friendly conditions. |
| Ad skepticism [13,97,98] | 10 | "Consumers would be better off" without advertising. |
| Black Friday attitude | 5 | Black Friday deals encourage unnecessary consumption. |
| Brand evaluation [13,75,99] | 9 | Unconvincing/convincing. |
| Sustainability evaluation [94] | 6 | Profit-oriented/charitable. |
| Sustainability skepticism [98] | 6 | This ad uses graphic elements to deceive consumers regarding the environmental impact of the product. |

### 3.1.3. Data Analysis

Statistical tests were performed using IBM SPSS 27 statistical software. Exploratory factor analyses based on maximum likelihood of all scales were performed. The factor loadings using the Promax rotation method showed values above the threshold of 0.5 recommended by Hair, Ringle, and Sarstedt [100]. We could confirm the discriminant validity of all constructs and items, as the items loaded—without significant cross-loadings—on their respective factors. The internal consistency reliability of the scales was assessed using Cronbach's alpha coefficients (Table 4). We removed one item each from the sustainabil-

ity consciousness scale and the sustainable fashion-related consciousness scale, as factor analysis revealed that they did not measure the intended constructs. Means and standard deviations were computed for all relevant variables. Pearson's correlation coefficients were computed between all relevant variables. Lastly, stepwise multiple regression analysis was used to examine which factors influence brand evaluation and sustainability evaluation.

**Table 4.** Descriptive statistics.

| | Variable | M | SD | α | 1 | 2 | 3 | 4 | 5 | 6 | 7 | 8 | 9 | 10 | 11 |
|---|---|---|---|---|---|---|---|---|---|---|---|---|---|---|---|
| 1 | Sustainability consciousness | 4.97 | 0.51 | 0.76 | | | | | | | | | | | |
| 2 | Ad skepticism | 4.21 | 0.52 | 0.69 | 0.19 * | | | | | | | | | | |
| 3 | Purchase behavior | 5.05 | 0.66 | 0.77 | 0.65 ** | 0.16 * | | | | | | | | | |
| 4 | Fashion sustainability consciousness | 5.29 | 0.68 | 0.78 | 0.38 ** | 0.37 ** | 0.29 ** | | | | | | | | |
| 5 | Fashion purchase behavior | 4.80 | 0.87 | 0.90 | 0.58 ** | 0.14 | 0.72 ** | 0.29 ** | | | | | | | |
| 6 | Attitude towards Black Friday | 4.58 | 0.79 | 0.75 | 0.31 ** | 0.13 | 0.29 ** | 0.19 * | 0.29 ** | | | | | | |
| 7 | Brand evaluation | 3.23 | 0.69 | 0.96 | −0.05 | 0.19 * | −0.10 | 0.07 | −0.06 | 0.24 ** | | | | | |
| 8 | Sustainability evaluation | 3.26 | 0.72 | 0.94 | −0.02 | 0.21 ** | −0.09 | 0.05 | −0.01 | 0.24 ** | 0.89 ** | | | | |
| 9 | Education | 3.56 | 0.69 | | 0.08 | −0.06 | 0.12 | −0.05 | 0.04 | 0.05 | 0.08 | 0.09 | | | |
| 10 | Income | 1.63 | 0.81 | | −0.08 | 0.03 | 0.01 | −0.29 ** | 0.07 | 0.09 | −0.08 | −0.07 | 0.09 | | |
| 11 | Gender | 1.85 | 0.38 | | 0.06 | 0.02 | 0.22 ** | 0.07 | 0.14 | −0.03 | −0.18 * | −0.14 | −0.10 | −0.11 | |
| 12 | Sustainability skepticism | 3.77 | 0.68 | 0.93 | 0.03 | 0.21 * | 0.04 | 0.14 | 0.08 | 0.20 * | 0.71 ** | 0.70 ** | −0.05 | −0.13 | −0.04 |

Note. * $p \leq 0.05$; ** $p \leq 0.01$. $n = 148$. Maximum value equals 6 for all variables except education (max = 4), income (max = 4), and gender (1 = male, 2 = female, 3 = diverse).

*3.2. Results, Part 2*

3.2.1. Participants

The majority of the participants were female (83.8 percent), the mean age was 26.42 years (SD = 5.47), and the ages ranged from 16 to 53 years. More than 90 percent were well-educated (at least high school graduate level) and 65.5 percent were highly educated (university degree level). The mean income was rather low; more than half had a monthly income of less than 1.500 €.

3.2.2. Descriptive Statistics

Table 4 shows means, standard deviations, internal consistencies, and intercorrelations of the study variables. The highest correlation coefficients with consumer evaluations were found between consumers' attitude towards Black Friday and (a) brand evaluation and (b) sustainability evaluation (both: r = 0.24, $p < 0.01$). Ad skepticism also correlated significantly with (a) brand evaluation (r = 0.19, $p < 0.05$) and (b) sustainability evaluation (r = 0.21, $p < 0.01$). The overall highest correlation coefficient was found between general purchase behavior and fashion purchase behavior (r = 0.72, $p < 0.01$).

3.2.3. Stepwise Multiple Regression

As shown in the model (Figure 1), the predictive ability of six variables (sustainability consciousness, fashion-related consciousness, purchase behavior and attitude towards sustainable products, fashion-related purchase behavior and attitude towards sustainable fashion, ad skepticism, and attitude towards Black Friday) on consumers' brand evaluation (Table 5) and on sustainability-related evaluation (Table 6) was tested in a stepwise multiple regression. Gender, income, and education level were included as control variables.

**Table 5.** Stepwise regression with brand evaluation as dependent variable.

| Variable | Regression Coefficient | $R^2$ | Beta Coefficient | T | F | *p*-Value |
|---|---|---|---|---|---|---|
| Attitude towards Black Friday | 0.24 | 0.06 | 0.28 | 3.38 | 8.77 | 0.00 |
| Purchase behavior | −0.23 | 0.09 | −0.22 | −2.63 | 7.15 | 0.01 |
| Ad skepticism | 0.25 | 0.13 | 0.19 | 2.40 | 6.83 | 0.02 |

**Table 6.** Stepwise regression with sustainability evaluation as dependent variable.

| Variable | Regression Coefficient | $R^2$ | Beta Coefficient | T | F | *p*-Value |
|---|---|---|---|---|---|---|
| Attitude towards Black Friday | 0.24 | 0.06 | 0.27 | 3.26 | 8.68 | 0.00 |
| Ad skepticism | 0.29 | 0.09 | 0.21 | 2.66 | 7.19 | 0.02 |
| Purchase behavior | −0.22 | 0.13 | −0.20 | −2.40 | 6.87 | 0.02 |

Assumptions of linear regression were tested. Normality of residuals was given in the sustainability evaluation model ($p > 0.05$), but the normality assumption was not met in the brand evaluation model ($p < 0.05$). However, according to Schmidt and Finan [101], violations of this assumption have no noticeable impact on the results in linear regressions. The homoscedasticity assumption was violated, but, according to Fox, this violation is only problematic when the "largest error variance is more than about 10 times the smallest" [102] (p. 307), which is not the case.

As regards brand evaluation, consumers' attitudes towards Black Friday, purchase behavior, and ad skepticism explained 12.5% of the variance. The model was statistically significant ($p < 0.05$).

Attitude towards Black Friday had the greatest predictive ability, explaining 5.7% of the variance. A negative standardized beta coefficient was found for purchase behavior, suggesting that those who had sustainable purchase behavior gave a low brand evaluation. Attitude towards Black Friday and ad skepticism presented positive beta coefficients, suggesting that those who had a negative attitude towards the sales event and showed high ad skepticism gave good brand evaluations.

With regard to consumers' evaluations of brands' sustainability-related performances, consumers' attitudes towards Black Friday, purchase behavior, and ad skepticism also explained 12.5% of the variance. The model was statistically significant ($p < 0.05$).

Again, a negative standardized beta coefficient was found for purchase behavior, suggesting that those who had sustainable purchase behavior gave a low sustainability evaluation. Attitude towards Black Friday and ad skepticism presented positive beta coefficients suggesting that those who had a negative attitude towards the sales event and showed high ad skepticism gave good sustainability evaluations. However, ad skepticism appears to have a stronger predictive ability than purchase behavior in this model (and vice versa for brand evaluation).

### 3.3. Discussion, Part 2

In the second part of the study, we investigated which factors are most likely to predict positive or negative consumer evaluations of Black Friday-related posts by sustainable brands. Data showed that three factors have predictive ability for brand evaluation as well as sustainability evaluation: Black Friday attitude, purchase behavior, and ad skepticism.

The most important predictive factor was consumers' attitude towards Black Friday. Those who were critical of conventional Black Friday campaigns due to their environmental and social impact were likely to give 'sustainable' Black Friday campaigns good evaluations, suggesting that they perceived the presented campaigns as more sustainable. Conversely, those who were not critical of Black Friday gave negative brand evaluations.

These consumers might prefer high discounts instead of making donations or receiving a small gift (or no incentive at all). These assumptions are supported by the fact that attitude towards Black Friday correlates significantly with sustainability involvement (i.e., sustainability consciousness and purchase behavior).

The Hypothesis (H1) that sustainability involvement (i.e., sustainability consciousness and purchase behavior) has a significant predictive ability is not supported by our data. While purchase behavior is a predictor, sustainability consciousness is not. High scores for purchase behavior (i.e., sustainable purchase habits and preference for sustainable products) predict negative brand and sustainability evaluations. These findings may be explained by the assumption that consumers who generally have a minimalist approach to shopping dislike sales events that encourage consumption. Siebenhüner's concept of homo sustinens, who is more likely to be intrinsically motivated than motivated through external (e.g., monetary) incentives, is also relevant in this regard.

Lastly, ad skepticism is a predictive factor for brand and sustainability evaluation: consumers who exhibited high ad skepticism gave 'sustainable' Black Friday campaigns good evaluations, which suggests that the ads were convincing. According to Matthes and Wonneberger [11], high informational utility leads to lower ad distrust. Thus, while consumers might generally be skeptical of ads, the high informational utility of the shown ads may render the ads trustworthy. In addition, Minton et al. [28] suggest that social media ads are potentially perceived as more credible than traditional media. The medium, therefore, may also have an effect on consumer evaluations.

Although the regression does not demonstrate a predictive ability of sustainability consciousness for consumer evaluation, there is a statistically significant positive correlation between sustainability consciousness and ad skepticism. There is also a positive correlation between ad skepticism and sustainable purchase behavior. Since sustainable purchase behavior predicts negative evaluations, while high ad skepticism predicts positive evaluations, there appears to be an additional mechanism or mediator that has an impact on this relationship.

The indication that participants who exhibited high ad skepticism gave positive evaluations might create the impression that they did not recognize the greenwashing and bluewashing strategies. However, there is a high positive correlation between the evaluations and sustainability skepticism. High sustainability skepticism suggests that participants recognized or at least suspected green- and/or bluewashing. However, we cannot rule out participant bias. Participants may have merely pretended to recognize greenwashing and/or bluewashing because they believed they were expected to. They may also have only recognized the strategies upon being questioned, meaning the question prompted them to take a closer look. In that case, however, we would expect to see a negative effect on all evaluations after the first set of posts, since the sustainability skepticism questions came after the first evaluation. We suspect, however, that the participants were able to recognize the green- and bluewashing, but because of the ads' high informational utility, which leads to lower ad distrust [11], they did not evaluate the ads negatively. This idea is supported by the fact that there is also a significant correlation between sustainability skepticism and ad skepticism; those who indicated occurrences of green- and/or bluewashing were generally skeptical of all ads (also green and 'blue' ads), which gives reason to assume that they subjected the ads to increased scrutiny. It is possible that they tolerate green- and bluewashing in ads by sustainable brands because they generally trust these brands and want to support their cause. According to Pirsch, Gupta, and Grau [103], long-term commitment to CSR programs provokes less consumer distrust than short-term promises. Hence, we can assume that sustainable brands, which are more likely to make long-term CSR commitments, are perceived as more trustworthy than conventional fashion brands.

## 4. Limitations

Some limitations should be considered in interpreting the results. Firstly, no data were collected regarding the proportion of greenwashed and/or bluewashed ads among all Black

Friday ads posted by sustainable brands. We can, therefore, neither make assumptions about the scale of the problem nor can we make generalizations for all Black Friday ads posted by sustainable brands. However, the study signifies a starting point for this vital field of research and offers a literature-based model of greenwashing and bluewashing categories that can be used for further research in this regard. The prevalence of the issue among sustainable brands should be the subject of future research.

One limitation of the second part of the study was the participant sample. Firstly, the sample size was rather small and, secondly, the majority of the participants were young, highly educated women, meaning that generalizations of the results may be problematic. This sampling bias is in part owed to the fact that women are more likely to show interest in topics that are relevant to this study, including fashion and sustainability issues [104–108]. Additionally, the link to participate in the survey was shared in sustainability-related groups on various social media platforms. Data suggest that women are overrepresented among sustainability-related Instagram influencers as well as their followers [109]. Furthermore, the study focused on Instagram and familiarity with the social media platform was a prerequisite for survey participation. Hence, a younger participant sample was to be expected. Nevertheless, we recognize that monetary incentives for participation might yield a more representative sample.

Furthermore, in order to determine participants' sustainability skepticism, we relied on participant reports (questionnaire responses). An experimental approach using manipulated material was beyond the scope of this study. Hence, we cannot make definitive assumptions about whether participants recognized greenwashing and/or bluewashing occurrences. Moreover, the stimuli exhibited combined strategies, meaning the greenwashing and bluewashing categories were not presented individually. Further research might investigate the effectiveness of the individual categories.

Nevertheless, the study provided some valuable insights into the factors influencing brand evaluation. These findings contribute to a better understanding of the appeal of green and 'blue' advertising and also constitute valuable data for sustainable fashion brands. Future research should explore the mediating factor for the correlation between purchase behavior and brand evaluation.

## 5. Implications

Our research demonstrates that a wide variety of greenwashing and bluewashing strategies is employed in Black Friday advertisements by sustainable fashion brands. Despite a possible temporary competitive advantage, sustainable SMEs need to recognize that the extensive use of greenwashing and bluewashing strategies in their social media advertisements legitimizes this practice by non-sustainable brands. Ultimately, the use of such deceptive tactics makes it more difficult for consumers to make sustainable purchase decisions and thus diminishes the competitive advantage of sustainable SMEs. Our findings also suggest that truly green consumers do not approve of the employed marketing tactics. Siebenhüner's concept of homo sustinens, who exhibits green behavior based on a sense of moral obligation and is less likely to respond to monetary incentives, is applicable to these consumers [71]. In addition, since previous data suggest that strong ESG scores reflect well in economic performance [110], companies that pay lip service to their sustainability policies risk financial damage. Hence, SMEs are strongly discouraged from participating in these practices so as not to risk credibility.

For the sake of sustainable SMEs as well as the consumer and the environment, it is imperative that the use of greenwashing and bluewashing is curbed. Currently, the sustainability market relies heavily on the self-commitment of businesses. However, without regulative measures, this approach is easily instrumentalized and misappropriated. While certification systems constitute some improvement in this respect, major policy changes and regulative measures are required in order to truly inhibit deceptive sustainability-related marketing tactics. Nonetheless, sustainable SMEs are well-advised to refrain from the misleading strategies frequently employed by conventional brands. Our greenwashing

and bluewashing model constitutes a valuable resource for both businesses and research in this field.

## 6. Conclusions

The competitiveness of the fashion market as well as the popularity of Black Friday among consumers may drive sustainable brands to participate in the sales event despite its negative environmental and social impacts. While perpetuating such unsustainable habits is already problematic, the more pressing concern is that sustainable brands use greenwashing and/or bluewashing strategies in their social media communications to deceive consumers in order to enjoy the financial benefits while still upholding their sustainable corporate images. These sustainable brands legitimize the use of greenwashing and bluewashing tactics and enable conventional brands to follow suit, diminishing the competitive advantage of sustainable marketing.

In the first part of the study, we demonstrated that 39 brands were guilty of at least one greenwashing and/or bluewashing strategy. A total of 115 strategies were used. Recorded strategies included the substitution of discounts with other offers (promotional gifts or donations), the use of misleading language (rebranding and the Sin of Vagueness), the use of imagery, the endorsement by credible celebrities, and the use of misleading claims (ethical consumerism claim, the Sin of Fibbing, and the Sin of the Lesser of Two Evils).

The gravity of this issue was demonstrated in the second part of the study, which showed that even those participants who indicated that they recognized green- and/or bluewashing instances in the ads shown (i.e., those who exhibited high sustainability skepticism) gave good evaluations regarding the brands and their sustainability performances. Evaluations also positively correlated with ad skepticism and attitude towards Black Friday. Additionally, the study demonstrated that consumers' negative attitudes towards Black Friday and high ad skepticism had statistically significant predictive abilities for positive brand evaluation and sustainability evaluation. Hence, we can assume that consumers who recognize the negative social and environmental impact of Black Friday prefer seemingly more sustainable alternatives over conventional Black Friday campaigns. Consumers who generally distrust ads appear to find these campaigns appealing, perhaps due to their high informational utility.

Sustainable purchase behavior is a predictive factor for negative brand evaluations, suggesting that consumers who shop responsibly disapprove of Black Friday campaigns by sustainable brands. These insights indicate that Black Friday campaigns by sustainable brands may be effective for consumers who are generally concerned about environmental and social sustainability issues, but not for those whose behavior is actually sustainable. On the basis of these findings, we advise sustainable SMEs to refrain from using greenwashing and bluewashing strategies in order to maintain credibility among sustainability-conscious consumers.

**Author Contributions:** Conceptualization, A.S.; methodology, A.S. and E.S.; formal analysis, A.S. and E.S.; investigation, A.S.; writing—original draft preparation, A.S., E.S. and H.W.; writing—review and editing, E.S. and H.W.; visualization, A.S.; supervision, H.W. All authors have read and agreed to the published version of the manuscript.

**Funding:** This research received no external funding. Open Access Funding by the University of Vienna.

**Data Availability Statement:** The data of this study are available from the corresponding author upon reasonable request.

**Conflicts of Interest:** The authors declare no conflict of interest.

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
