# Peer review of "Greenwashing and Bluewashing in Black Friday-Related Sustainable Fashion Marketing on Instagram"

_sustainability, doi:10.3390/su14031494_

Round 1

Reviewer 1 Report

Dear authors,

Thank you for interesting topic you covered in your research. As sustainability fashion is gaining interest not just in general public, but also among researchers, your paper is a good expansion to body of knowledge related to consumer behavior in sustainability. Use of two different approaches in your research is adding the credibility to your conclusions. Although manuscript is adequately formulated and referenced still some suggestions are offered to improve your paper.

At the end of the Introduction section please identify shortly contributions of your paper. What are your main contributions to the theory.

It is suggested to formally state hypotheses in the manuscript, not just to present conceptual model. Furthermore, conceptual model is suggested to be placed after the hypotheses are posited (preferably in the Literature review section). Preferably place conceptual model at the end of Literature review section.

Authors mentioned in Table 3 that they used 5 items for Black Friday attitude but not mentioning the reference for those items. Please add the reference based on what you have formed your Black Friday attitude scale.

Please add what method have you used for factor analysis (pg13 ln544-545).

Purchase behavior is negative in both regressions. What have you assumed in hypothesis? As they were not formally stated (maybe it would be better they have been). So, please add hypotheses in the literature review and if hypotheses related to purchase behavior is positive (and your results appear negative, although statistically significant) please refer to that in the discussion section.

In Conclusion section please reconsider to add Implications for practice/Managerial implications.

Reviewer 2 Report

  • You need to present your method in a more cohesive way somewhere in the paper so it is easier to grasp the full extent of the different method used
  • the references need to be update

Reviewer 3 Report

I am not familiar with the methods used for the analysis and cannot fully understand the method and assess the analysis. My impression is that things are correct and that the authors did a good job.

Here are some suggestions to provide:

  1. The researchers conducted a full literature review. I was not aware of the terms "Greenwashing" or "Bluewashing" before reading this article, and I was aware of "Greenwashing" after reading this literature review, but as the authors pointed out, "Bluewashing" was a new phenomenon and lacked standardized terminology, Therefore, I think the author should give a definition under your research framework in chapter 1.1.3.

2.Line554- most of the participants are young women, please explain what the sample represents. Why did you choose these samples?

  1. I suggest adding two chapters of management significance and theoretical contribution. The existing conclusions are not sufficient.

  1. I suggest expanding your research limits. I think you should separate the research limits part of the discussion into a separate chapter.
